# Validation of Garmin HRM-Pro for Assessment of Spatiotemporal Parameters During Treadmill Running: Agreement with Three Motion Analysis Systems

**DOI:** 10.3390/s25175407

**Published:** 2025-09-02

**Authors:** José Carlos Cabrera-Linares, Cristian Martínez Salazar, Juan Antonio Párraga Montilla, Pedro Ángel Latorre Román

**Affiliations:** 1Departamento de la Expresión Musical, Plástica y Corporal, Universidad de Jaén, 23071 Jaén, Spain; jccabrer@ujaen.es (J.C.C.-L.); platorre@ujaen.es (P.Á.L.R.); 2Departamento de Educación Física, Deportes y Recreación, Universidad de La Frontera, Temuco 1145, Chile; cristian.martinez.s@ufrontera.cl

**Keywords:** running, wearable, spatiotemporal parameters, validity

## Abstract

Background: The aim of this study was to validate the accuracy of the spatiotemporal parameters of running provided by the Garmin HRM-Pro band through a concurrent comparison with the OptoGait system (version v.1.14.11), the Stryd power meter, and 2D photogrammetric analysis in recreational runners at different paces (9 and 12 km/h). Methods: Fifty recreational runners (mean age: 22.14 ± 2.71 years) engaged in this study. Participants ran on a treadmill for 2 min at speeds of 9 and 12 km/h. Spatiotemporal parameters (ground contact time, step length, flight time, cadence, and vertical oscillation of center of gravity) were recorded using (1) the Garmin HRM-Pro band; (2) the OptoGait system; (3) the Stryd power meter; and (4) 2D photogrammetric analysis. Results: Only in relation to the VO of the center of gravity does the Garmin device yield higher values (*p* = 0.006) compared to video analysis. At 12 km/h, significant differences were found between devices in all the analyzed variables. In comparison to video analysis, the Garmin device does not show significant differences in any variable at this speed. The relative reliability parameters of the Garmin device at two different speeds showed excellent values across all analyzed variables. The Bland–Altman plots showed appropriate limits of agreement indicating good agreement between devices. Conclusions: The Garmin HRM-Pro Band can be used for the assessment of spatiotemporal variables of running in sports science, clinical gait assessment, and training optimization.

## 1. Introduction

Today, running is one of the most practiced sports in the world [1], with thousands of runners (from amateur to elite) taking part every weekend in running races of different distances worldwide [2]. Running offers the athlete health benefits, improving musculoskeletal and cardiovascular health and having a beneficial effect on body mass, body fat, resting heart rate, VO_2_max, triglycerides and HDL cholesterol, and psychological state [3,4]. Its popularity is due, among other factors, to the satisfaction of physical and psychological health needs, goal achievement, tangible rewards, social influences, and easy availability [5]. Achievement motives, such as competition with other runners and the attainment of personal goals, are also important, and, in this regard, improving one’s personal best is a key performance motivation for all runners during a race. In this context, there are several factors that can predict endurance running performance. These include physiological variables (lactate threshold, maximum oxygen uptake (VO_2_max) [6]) and biomechanical factors linked with mechanical efficiency and running economy [7]. In particular, the latter can be modified through several variables, and in this regard, some running kinematic factors appear to benefit running economy, including a self-selected stride length within a 3% shorter range, lower vertical oscillation (VO) of center of gravity, reduced leg extension at toe-off, greater stride angles, and short ground contact times (GCTs) [8].

Nonetheless, more than half of the adult running population sustains an injury after covering 1000 km [9]. The spatiotemporal parameters of running not only influence performance but are also linked to injury risk, which makes running technique very important for the prevention of injuries [10]. A recent systematic review showed that runners with a history of injuries tend to exhibit greater variability in spatiotemporal parameters, which could indicate a compensatory pattern or persistent risk [11]. Notably, overstriding, by increasing the impact peak force, poses a risk for patellofemoral pain syndrome [12]. Moreover, a prolonged GCT is linked with lower leg pain [13]; thus, the duty factor (defined as the proportion of GCT relative to the total stride time) is an injury risk factor [14,15]. Additionally, the duty factor has also been associated with running performance and economy [16,17], and, in this regard, changes in running technique can influence running economy and lead to improved running performance [18]. Various studies [19,20,21] have used different forms of retraining to modify specific spatiotemporal running variables as a strategy for preventing injury and improving performance.

Therefore, the analysis, evaluation, and monitoring of spatiotemporal running variables is important in terms of injury prevention and enhancing running performance. The gold standard in kinematic running analysis has been 3D motion capture system using high-speed cameras [22]. Nevertheless, this system has some limitations, including being time-consuming, its high cost, and the need for close contact with athletes, which is uncomfortable for them and limits the user’s performance. In addition, some athletes may consider it to be intrusive, which can result in the modification of the naturalness of their movement patterns. Also, this system has limited capture volume, which is the key factor that limits the capability of optical motion capture for long-distance running [23].

Likewise, optical systems have also facilitated the recording of these variables, even in field situations—the OptoGait system, in particular, has been used previously to assess spatiotemporal running parameter at different velocities [24]. More recently, an alternative for measuring spatiotemporal parameters could be inertial sensors (small devices that include accelerometers, gyroscopes, or magnetometers) since this technology can be considered as a low-cost system, with lightweight devices that are easy to carry and offer unlimited investigation of movement [25]. In this regard, wearable technology (including those devices that are worn directly on, or loosely attached to, a person) has been growing constantly in popularity worldwide since 2016 to become the number one trend in a 2025 survey [26]. Specifically, wearable technology has quickly become an effective tool to monitor running biomechanics and creates an opportunity to combine this information with training performance recommendations [27], to quantify running intensity and aerobic fitness [28,29], as well as for injury prevention strategies [30]. Consequently, more than 90% of recreational runners utilize a wearable device to know their running metrics, such as GCT, step length (SL), flight time (FT), cadence, and VO of the center of gravity [31]. Hence, as more people are using wearable technology (for recreational purposes, for experienced athletes, and research), there is a need to establish the validity of these devices [1].

Previous studies have demonstrated that wearables are valid and reliable devices for measuring spatiotemporal variables during running [1,29,32]. However, these studies present significant limitations, such as variability in measurements between devices and a lack of consensus on statistical methods and validity criteria. The rapid advancement of wearable technology in sports science has led to a growing demand for reliable, valid, and standardized measurement tools. However, the heterogeneity of validation protocols and the limited testing across diverse populations pose significant challenges to the clinical and athletic applicability of these devices. Ensuring that the produced data are both accurate and practically meaningful is crucial for athletes, coaches, and researchers. In response to this need, the present study introduces a novel methodological approach by performing a synchronous validation of three widely used commercial systems (Stryd, OptoGait, and 2D video analysis) under controlled laboratory conditions and at two distinct running speeds. This dual-speed, multi-device comparison not only enables the assessment of inter-device agreement but also facilitates the identification of potential measurement biases associated with running velocity, an issue that remains underexplored in the current literature.

The Garmin HRM-Pro band has been chosen in this study to compare with other devices due to its increasing adoption among recreational and professional runners [31], its ability to measure multiple spatiotemporal parameters (GCT, SL, FT, cadence, and VO) non-invasively, and the need to validate its metrics against reference systems. Unlike devices such as the Stryd, which require specific calibration [33], the HRM-Pro band offers an integrated solution with popular platforms such as Garmin Connect (v.7.25.0), increasing its practical applicability in daily training [30]. However, the current literature lacks studies that directly compare its accuracy with standardized systems (OptoGait) and 2D video analysis under real-life running conditions [1,31]. This tripartite approach (OptoGait, Stryd, and 2D video analysis) will allow us to assess the device’s validity and establish its specific limitations for capturing different spatiotemporal parameters. Therefore, the aim of this study was to validate the accuracy of spatiotemporal parameters (GCT, SL, FT, cadence, and VO) provided by the Garmin HRM-Pro band by concurrent comparison with the OptoGait system, the Stryd power meter, and 2D photogrammetric analysis in recreational runners at different paces (9 and 12 km/h).

## 2. Materials and Methods

A priori sample size estimation for a repeated-measures ANOVA was conducted using G*Power 3.1. The analysis was based on a within-subjects design involving four measurement conditions (Garmin, Stryd, Optogait, and a camera-based system). Assuming a medium effect size (Cohen’s f = 0.25), a significance level of α = 0.05, statistical power of 0.80, and a moderate correlation among repeated measures (r ≈ 0.5), the minimum required number of subjects was 28. This study included 50 participants, each measured under all four conditions, exceeding the minimum requirement and ensuring sufficient power to detect meaningful within-subject differences across devices. Therefore, this cross-sectional study involved a total of 50 healthy adults (mean age: 22.14 ± 2.71 years; BMI: 23.82 ± 2.56 kg/m^2^; 18 women). The participants were recruited from the University of Jaén (Andalusia, Spain). The inclusion criteria were as follows: (1) is over 18 years old; (2) does not have any injury that prevents performing physical activity; (3) has not performed intense efforts 72 h before the evaluation; (4) has not taken any drugs that improve sports performance; (5) has the ability to run stably on a treadmill at different paces. All the participants involved in this study signed an informed consent before starting the assessment. This study was conducted following the norms of the Declaration of Helsinki [34]. In addition, the European Union guidelines on Good Clinical Practice (111/3976/88 of July 1990), specified in a national legal framework for clinical research in humans (Royal Decree 561/1993 on clinical trials), were followed. This study was also approved by the Ethics Committee of the University of Jaén (Spain) with reference code OCT.20/7.PRY.

### 2.1. Anthropometric Data

Body mass was registered using a weight scale (Seca 899, Hamburg, Germany), and body height was measured with a stadiometer (Seca 222, Hamburg, Germany).

### 2.2. Running Analysis

In the current study, GCT, SL, FT, cadence, and VO were registered during a running test at different speeds (9 km/h and 12 km/h) on a treadmill. To assess and compare the data, four different devices were used:

Running dynamic data were recorded with the Garmin Forerunner 635 smartwatch paired with a heart rate band (HRM-Pro Run; Garmin Ltd., Olathe, KS, USA). The smartwatch was placed on the left wrist and the HRM-pro band on the trunk at the xiphoid process, following the manufacturer’s instructions.

### 2.3. Devices for Comparison

Stryd (Stryd Power meter, Stryd Inc., Boulder, CO, USA): Stryd^™^ is a carbon fiber-reinforced foot pod (attached to the shoe) that weighs 9.1 g. Based on a 6-axis inertial motion sensor (3-axis gyroscope, 3-axis accelerometer), the device stores data at 1 Hz sampling frequency. This device has been validated previously [35]. The firmware version used was 1.1.9 and data were extracted from the Stryd application (https://www.stryd.com/powercenter/athletes/b0b9bcdd-d5a4-5c14-6f90-0c695fe15030/profile?sid=b0b9bcdd-d5a4-5c14-6f90-0c695fe15030, accessed on 30 January 2025). Notice that Stryd does not require traditional manual calibration. However, it is important to follow these steps: (a) pair properly with the Stryd app or a compatible watch; (b) place the pod in the center of your shoelace, securely fastened; (c) Stryd will self-calibrate using GPS or accelerometry data; and (d) ensure it is fully charged and firmware is up to date before each test.

OptoGait system (OptoGait^®^ Microgate, Bolzano, Italia): OptoGait is an optical data acquisition system composed of a transmitter and a receiver bar. Each 1 m bar contains 96 infrared LEDs (1.041 cm resolution) and is located on the transmitter bar, continuously communicating with the LEDs located on the receiver bar. The bars measure flight and contact times during execution with an accuracy of 1/1000 of a second. OptoGait has been previously validated for the assessment of spatiotemporal gait parameters in adults [36]. Two parallel bars were placed alongside each other on the side of the treadmill at the same level as the contact surface. Before initiating the assessment, and to ensure proper calibration of the device, the following steps must be completed. (a) Align the transmitter and receiver bars in parallel, as specified by the manufacturer. (b) Connect both bars to the computer via USB or Bluetooth and launch the OptoGait software. (c) Perform a signal integrity check to confirm that all LEDs are active. (d) Calibrate the sensor height to align with floor level. (e) Configure the software by selecting the activity mode (running), activating the required number of sensors, and setting the appropriate sampling frequency.

Video analysis: A high-resolution video camera (Casio Exilim EX-10, Casio, Tokyo, Japan) was used to register running parameters. Prior to recording, standardized calibration procedures were implemented: (a) A camera was mounted on a stabilized tripod at 5 m distance and 1 m height, with perpendicular alignment. (b) Define the calibration plane by placing a reference object of known dimensions at ground level within the recording area; in this regard, a 1 m calibration rod (accuracy ± 1 mm) was positioned parallel to the running direction. (c) Ensure that the entire calibration object is visible in the camera frame and aligned with the motion path; in this regard, the full calibration object remained visible in-frame under controlled lighting (1000 lux) to maximize contrast. (d) If required by the software, set the scale by assigning real-world values to the reference points. (e) Confirm proper lighting and image sharpness to maximize the contrast between the markers and the background. Recordings were made at 120 fps with shutter speed > 1/1000 s (motion blur < 0.5 pixels), capturing 120 s of steady-state running at 9 and 12 km/h. Video analysis was performed using the open-source software Kinovea (version 2023.1.2). A trained member of the research team conducted the analysis, following prior standardization for assessing spatiotemporal gait parameters since the standardization of protocols and training of evaluators are key to maximizing the reliability of 2D analysis [37]. Furthermore, previous studies support the use of 2D analysis for the assessment of spatiotemporal variables and basic running kinematics, especially in the sagittal plane, where intra- and inter-rater reliability is generally high and differences with respect to 3D analysis are small and clinically acceptable [37,38,39]. Hence, 2D analysis is an accessible and practical tool for clinical and field evaluation, allowing monitoring of spatiotemporal variables and rapid feedback to athletes and patients [39].

### 2.4. Procedure

The research team placed the Garmin 635 on the left wrist, and the HRM pro band was placed at chest on the trunk at the xiphoid process (according to the manufacturer’s instructions). Moreover, the Stryd device was attached to the shoelace of the right leg according to the manufacturer’s instructions. Each OptoGait bar was placed in parallel on the treadmill at the same level as the contact surface, and the camera was placed 5 m from the treadmill (laterally) and 1 m from the ground floor. Subsequently, a warm-up consisting of jogging for 3 min at the same speed was used in the assessment (i.e., jogging 3 min at 9 km/h and 12 km/h). The warm-up was conducted on another treadmill in the laboratory (the same model as the treadmill used in the evaluation). After that, the participant moved to the treadmill where the OptoGait and the video camera were placed. Once the participant was ready to start, the velocity of the treadmill was increased until it reached 9 km/h, and was maintained at that speed for 2 min. The same protocol was followed at the speed of 12 km/h. The treadmill inclination was zero at both speeds, and both protocols were randomized. A total recovery was allowed between protocols to be sure that the participants performed the test correctly, and to avoid fatigue (Figure 1). Although 120 s were recorded, the first and last 15 s were removed to prevent data from being skewed due to the participant’s acceleration and deceleration. Hence, 90 s were analyzed.

### 2.5. Statistical Analysis

Data were analyzed using SPSS v.22.0 for Windows (SPSS Inc., Chicago, IL, USA). Descriptive data are reported in terms of means and standard deviations (SDs) and percentage (%). Tests of normal distribution and homogeneity (Kolmogorov–Smirnov and Levene’s test, respectively) were conducted on all data periods after the analyses. To analyze the differences between devices in the various spatiotemporal variables, a repeated-measures analysis of variance (ANOVA) was conducted. To assess agreement between devices (relative reliability), the intraclass correlation coefficient (ICC) based on a two-way mixed-effects model with absolute agreement was used. Two forms of ICC were calculated: ICC (2,1), which evaluates the reliability of individual measurements obtained from different systems, and ICC (2,k), which evaluates the reliability of the mean values per subject, where k is the number of devices compared. This approach allows for the assessment of both the pointwise agreement between measurements and the overall stability of aggregated subject-level values. The analysis was performed for each of the biomechanical variables recorded by the devices (GCT, SL, FT, cadence, and VO). The ICC values were interpreted according to the criteria proposed by Koo and Li [40], where values below 0.5 are considered indicative of poor reliability, those between 0.5 and 0.75 indicate moderate reliability, those between 0.75 and 0.9 indicate good reliability, and those above 0.9 indicate excellent reliability. To evaluate the presence of heteroscedasticity—that is, error variability proportional to the magnitude of the measurement—Pearson’s correlation was calculated between the mean values of the two devices and the absolute differences between their measurements, following the recommendations of Bland and Altman [41] and Giavarina [42]. Additionally, Bland–Altman plots were generated to visually inspect potential systematic patterns in the distribution of measurement errors. The mean difference (bias) and its 95% confidence interval (CI) were used to determine the presence and significance of systematic bias. Additionally, a Pearson correlation analysis was employed. The magnitude of correlation between measurement variables was designated as follows: <0.1 (trivial), 0.1–0.3 (small), 0.3–0.5 (moderate), 0.5–0.7 (large), 0.7–0.9 (very large), and 0.9–1.0 (almost perfect) [38]. The significance level was set at *p* < 0.05.

## 3. Results

The analysis of spatiotemporal parameters using repeated-measures ANOVA revealed significant differences between measurement systems at both running speeds (9 km/h and 12 km/h) (Table 1). At 9 km/h, a significant difference was observed in SL between Stryd and Optogait (*p* < 0.05), with Stryd reporting lower values. Regarding FT, Garmin recorded significantly lower values compared to Stryd (*p* < 0.01), while Stryd exhibited significantly higher values than both Optogait (*p* < 0.01) and 2D video analysis (*p* < 0.05). In relation VO, Garmin showed significantly higher values than Stryd (*p* < 0.001), and Stryd presented significantly lower values compared to 2D video analysis (*p* < 0.05). Additionally, significant differences were found between Garmin and 2D video analysis (*p* < 0.01).

At 12 km/h, significant differences in cadence were found between Garmin and Stryd (*p* < 0.05), as well as between Stryd and Optogait (*p* < 0.05), with Stryd reporting lower values in both cases. Regarding SL, Stryd reported significantly lower values compared to Garmin (*p* < 0.001), Optogait (*p* < 0.001), and 2D video analysis (*p* < 0.01). In terms of FT, Garmin showed significantly lower values than Stryd (*p* < 0.001), while Stryd reported significantly higher values compared to Optogait (*p* < 0.001) and 2D video analysis (*p* < 0.001). For CGT, Garmin reported significantly higher values than Optogait (*p* < 0.05). Regarding VO, Garmin showed significantly higher values than Stryd (*p* < 0.001), and Stryd presented significantly lower values compared to 2D video analysis (*p* < 0.01). Significant differences were also observed between Garmin and 2D video analysis (*p* < 0.01).

The relative reliability parameters and agreement among the Garmin device and Stryd, Optogait, and 2D video analysis were assessed using intraclass correlation coefficients (ICC), Pearson’s correlation analysis (Table 2), and Bland–Altman plots (Figure 2, Figure 3, Figure 4, Figure 5 and Figure 6).

At 9 km/h, the ICC (2,1) values (reflecting the reliability of individual measurements) and the ICC (2,k) values (representing the reliability of averaged measurements) demonstrated excellent agreement for most variables. Regarding the comparison between Garmin and Stryd, ICC (2,k) values exceeded 0.99 for all variables: cadence (ICC = 0.995, 95% CI: 0.982–0.994), SL (ICC = 0.995, 95% CI: 0.982–0.994), FT (ICC = 1.0, 95% CI: 1.0–1.0), and CGT (ICC = 0.999, 95% CI: 0.996–0.999). VO also showed high consistency (ICC = 0.991, 95% CI: 0.976–0.992). For Garmin vs. Optogait, reliability was also excellent: cadence (ICC = 0.997), SL (ICC = 0.999), FT (ICC = 1.0), and CGT (ICC = 0.975). When Garmin is compared with the 2D video analysis, ICC (2,k) values were consistently high across all variables: cadence (ICC = 0.998), SL (ICC = 0.998), FT (ICC = 1.0), CT (ICC = 1.0), and VO (ICC = 0.997). In relation with Pearson’s correlation coefficients at 9 km/h, the results further supported a strong linear relationship between Garmin and the other systems. Garmin and Stryd showed very high correlations for cadence (r = 0.94, *p* < 0.001), SL (r = 0.92, *p* < 0.001), and FT (r = 0.95, *p* < 0.001). VO also showed a high correlation (r = 0.91, *p* < 0.001). Compared to Optogait, correlations were high for CGT (r = 0.93, *p* < 0.001) and moderate to high for FT (r = 0.84, *p* < 0.001). Notice that when compared to the 2D video analysis, correlation was excellent across all metrics: SL (r = 0.96, *p* < 0.001), FT (r = 0.97, *p* < 0.001), GCT (r = 0.98, *p* < 0.001), and VO (r = 0.89, *p* < 0.001).

At 12 km/h, overall, ICC (2,1) values (individual measurement reliability) and ICC (2,k) values (average subject-level reliability) showed moderate to excellent reliability across most variables. For instance, the comparison between Garmin and Stryd showed ICC (2,k) values above 0.90 for cadence (ICC = 0.93, 95% CI: 0.86–0.97) and CGT (ICC = 0.92, 95% CI: 0.85–0.96), indicating excellent consistency between devices. In contrast, reliability with the 2D video analysis system was more variable, especially for VO, which had an ICC (2,1) = 0.61 (95% CI: 0.42–0.78), suggesting moderate agreement. Pearson’s correlation coefficients between Garmin and Stryd were high for cadence (r = 0.94, *p* < 0.001) and SL (r = 0.87, *p* < 0.001), supporting a strong linear association between the devices. With Optogait, the correlation was very high for CGT (r = 0.95, *p* < 0.001) and moderate for FT (r = 0.68, *p* = 0.002). When compared to the camera-based system, a strong correlation was found for SL (r = 0.88, *p* < 0.001), but it was lower for VO (r = 0.60, *p* = 0.01).

Figure 2, Figure 3, Figure 4, Figure 5 and Figure 6 (9 km/h) and 7–11 (12 km/h) shows the Bland–Altman plots comparing the Garmin device with the other devices across the different spatiotemporal running variables analyzed.

At 9 km/h, the results of Bland–Altman plots showed minimal bias and narrow limits of agreement, thereby supporting the interchangeability between the devices. For Garmin vs. Stryd, bias was low for cadence (−1.2 ppm), with limits of agreement from −5.1 to +2.7 ppm. FT showed a bias of −2.6 ms (−9.2 to +4.0 ms), and GCT, 1.1 ms. In addition, VO showed a bias of (0.4 cm). In the case of Garmin vs. Optogait, the largest bias was observed in FT (−5.8 ms), though still within reasonable limits (−13.5 to +1.9 ms), whereas cadence (−1.0 ppm) and CGT (−1.3 ms) showed smaller differences. When compared to the 2D video analysis, biases were minimal across the board: −0.9 ppm for cadence, −0.5 cm for SL, −1.2 ms for FT, and −0.4 ms for GCT, while VO showed a small bias of −0.6 cm.

At 12 km/h (Figure 7, Figure 8, Figure 9, Figure 10 and Figure 11), Bland–Altman plots results indicated that the bias between Garmin and Stryd was small for Cadence (−1.2 steps/min), with limits of agreement between −5.1 and +2.7, suggesting good precision. For FT, the bias was −2.6 ms (−9.2 to +4.0), indicating a slight underestimation by Garmin. In the case of Optogait, larger biases were observed in FT (−5.8 ms), although the limits were relatively narrow (−13.5 to +1.9 ms), and it was minimal in CGT (−1.3 ms). For the 2D video analysis, the bias was notable in VO (−4.6 mm).

In addition, to assess whether the variability in measurement differences changed across the measurement range, heteroscedasticity was evaluated for each Bland–Altman plot using the correlation between the absolute differences and the means of each device pair. At both 9 km/h and 12 km/h, most comparisons did not show significant evidence of heteroscedasticity (*p* > 0.05), indicating that the measurement error remained consistent across the range of values. An exception was observed in cadence at 9 km/h between Garmin and Stryd, where a significant positive correlation (r = 0.397, *p* = 0.004) suggested potential heteroscedasticity. This implies that the measurement differences between these two devices may increase with higher cadence values. However, for the same variable at 12 km/h, no significant heteroscedasticity was detected, which reinforces the stability of measurement differences under faster running conditions. Overall, these results support the assumption of homoscedasticity in most device comparisons, validating the reliability of the Bland–Altman agreement assessments across the studied running speeds.

## 4. Discussion

The aim of this study was to validate the accuracy of spatiotemporal parameters (GCT, SL, FT, cadence, and VO) provided by the Garmin HRM-Pro band by concurrent comparison with the OptoGait system, the Stryd power meter, and 2D photogrammetric analysis in recreational runners at different paces (9 and 12 km/h). The main findings in this study were that, at both speeds, the Garmin HRM-Pro demonstrates excellent relative reliability parameters in measuring the analyzed variables, showing strong agreement with all the devices that were compared.

In addition to these results, one of the key methodological contributions of this study is the simultaneous validation of three widely used systems (Stryd, OptoGait, and 2D video analysis) under identical experimental conditions. To our knowledge, this is the first piece of research to conduct a synchronous comparison across these devices, which enables a direct assessment of inter-device agreement and improves the validity of the findings. This integrated validation approach offers a valuable framework for future studies seeking to evaluate the consistency of multi-sensor data in the context of running biomechanics. The present analysis shows that the Garmin HRM-Pro exhibits small and non-systematic biases when compared to the reference systems. Overall, good agreement was observed with inertial-based devices such as Stryd, while greater variability was found in comparisons with optical systems such as the video camera and OptoGait, particularly at higher running speeds (12 km/h). The overestimation of vertical oscillation compared to video analysis is consistent with previous studies linking chest-worn wearable devices to inflated measurements of the center of mass [32]. Although some increases in variability and bias were noted in specific comparisons—such as step length and flight time against the video camera—the direction of the bias was not clearly defined, preventing any firm conclusion regarding consistent over- or underestimation by the Garmin device. The absence of heteroscedasticity (*p* > 0.05) across all variables supports the consistency of measurement error across the observed ranges. However, García Pinillos et al. [35] after analyzing two different inertial sensors, found that the Stryd™ system underestimated contact time (CT) by 5.2% (*p* < 0.001) and overestimated flight time (FT) by 15.1% (*p* < 0.001) compared to video analysis (VA), whereas the RunScribe™ system underestimated CT by 2.3% (*p* = 0.009).The small discrepancies we observed reflect inherent limitations of inertial sensors under dynamic conditions [43]. These findings suggest that the Garmin HRM-Pro is a valid tool for estimating running kinematic parameters under treadmill conditions, particularly when compared with other inertial-based sensors. However, potential limitations should be considered when interpreting results against optical systems at higher speeds.

The results in the current study are in line with those presented in the systematic review conducted by Prisco et al. [44], since those authors concluded that commercial wearables show high reliability for spatiotemporal parameters such as cadence and SL, especially when validated against optical systems. In addition, Zeng et al. [45] conducted a research where they analyzed the validity and reliability of wearables equipped with inertial measurement units, concluding that these devices show moderate to excellent correlation with the gold standard. The current research complements previous findings by demonstrating that this accuracy is maintained, even when compared to reference systems such as 2D camera analysis and not just other wearables.

In relation to VO, these findings are consistent with the study by Smith et al. [32]. The authors compared four wearables (the Incus Nova, Garmin Heart Rate Monitor-Pro, Garmin Running Dynamics Pod, and Stryd Running Power Meter Footpod) against video analysis to assess the validity of these devices in measuring VO across varying running speeds (8–12 km/h), concluding that the wearable devices are valid and reliable tools for measuring VO during running. Similarly, Watari et al. [46] evaluated VO and GCT in 22 semi-elite runners at different paces (9.72, 10.8, 11.88, 12.96, 14.04 km/h) with zero inclination on a treadmill. The authors compared a commercial wearable (Garmin Forerunner 620) with 3D kinematics analysis, finding that both devices showed an excellent agreement; hence, the wearable is valid and accurate for the measuring of VO. However, Andersen and Nielsen [47] assessed the validity of a wearable (Garmin Forerunner 735xt) to measure VO at different speeds (10, 12 and 14 km/h) on a straight paved track. The authors found that the wearable is a reliable and valid device for measuring running dynamics in-field, but the results should be interpreted with caution since they systematically overestimate VO. The discrepancies among studies could be due to the fact that the authors registered metric data using a chest band (HRM-Run) that was older and had lower accuracy than the chest band used in the current study (HRM-Pro band).

With regard to GCT, our results align with the study by García-Pinillos et al. [35]. These authors studied the validity of spatiotemporal gait parameters (GCT, SL, FT, cadence) in comparison with some wearables (Stryd and RunScribe) and high-speed video analysis, reporting minimal errors between devices under steady-state running conditions. Similarly, Wundersitz et al. [48] concluded that GCT shows particularly high agreement between wearables and reference systems when running speed is controlled. In contrast, Koska et al. [49] demonstrated in their study that wearables have limitations in temporal parameters such as GCT during high-speed running. The differences across studies may be attributed to differences in sensor placement or the development and use of algorithms in newer-generation wearables [49].

### 4.1. Limitations and Strength

This study has some limitations that need to be mentioned: (1) The effect of fatigue was not assessed, and this could significantly alter movement patterns and, consequently, the accuracy of measurements. (2) The HRM Pro Band is usually used by the athletes outside the laboratory; thus, using a treadmill might reduce the ecological validity of the current study. (3) Inertial sensors placed on the lower limbs (e.g., ankle or shin) were not used in this study; the authors recommend that these be used in future studies to more accurately validate metrics derived from these devices and reduce the impact of errors induced by torso motion. Future research is encouraged to incorporate lower-limb IMUs to reduce the influence of upper body motion and enhance the validity of the data collected from wearable systems. (4) Although sensor placement was standardized by the same experienced researcher to ensure consistency, individual variability in stride length was not directly controlled, which may have influenced the accuracy of gait parameter measurements despite the use of a standardized treadmill protocol. Nevertheless, a strength of our study is that different running metrics provided by the HRM-Pro Band were compared against three different devices that have been previously validated.

### 4.2. Practical Applications

The present findings indicate that the Garmin HRM-Pro band exhibits excellent concurrent validity when benchmarked against established reference systems. This validation underpins a range of practical applications within the domains of performance monitoring and athletic training. In sports science, it offers a portable and accurate tool for monitoring running biomechanics during training, enabling performance tracking and technique adjustment. In clinical gait assessment, its agreement with reference systems suggests it can aid in preliminary evaluations or follow-up in rehabilitation, especially when laboratory equipment is unavailable. Finally, for training optimization, the device enables longitudinal monitoring of gait variables, allowing practitioners to make informed decisions to enhance performance and reduce injury risk.

## 5. Conclusions

To sum up, the Garmin HRM-Pro band showed excellent concurrent validity across key biomechanical variables at 9 km/h and 12 km/h, when compared to other devices. Therefore, Garmin HRM-Pro Band can be confidently used for the assessment of running biomechanics. Its agreement with reference systems across multiple gait variables supports its application in sports science, clinical gait assessment, and training optimization.

## Figures and Tables

**Figure 1 sensors-25-05407-f001:**
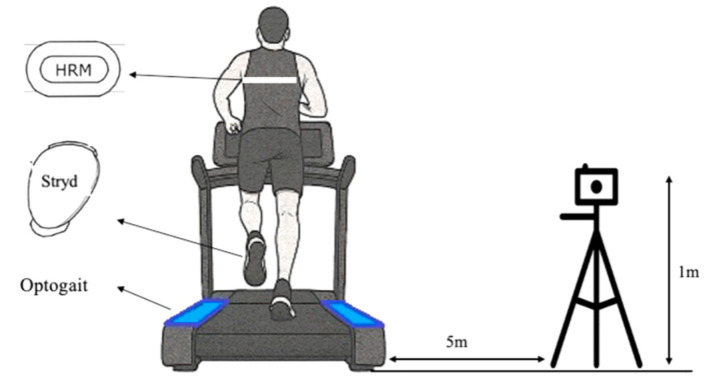
Protocol study and wearables placed on the participant’s body during assessment.

**Figure 2 sensors-25-05407-f002:**
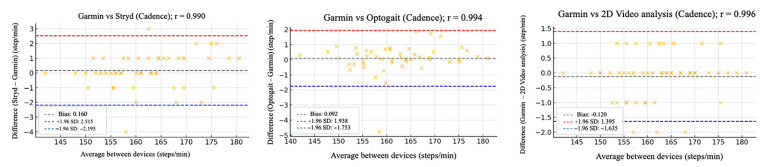
Bland–Altman plots comparing cadence between the Garmin and the other devices at 9 km/h.

**Figure 3 sensors-25-05407-f003:**
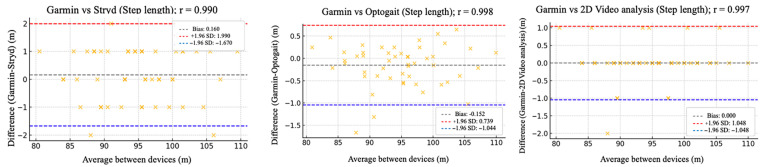
Bland–Altman plots comparing step length between the Garmin and the other devices at 9 km/h.

**Figure 4 sensors-25-05407-f004:**
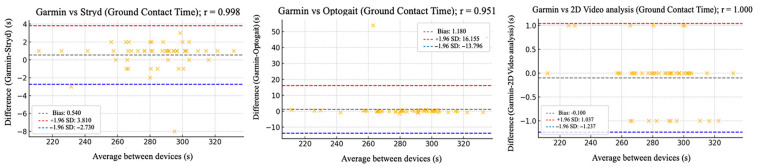
Bland–Altman plots comparing ground contact time between the Garmin and the other devices at 9 km/h.

**Figure 5 sensors-25-05407-f005:**
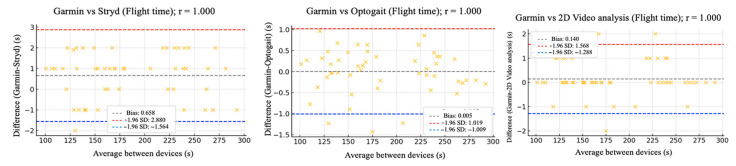
Bland–Altman plots comparing flight time between the Garmin and the other devices at both 9 km/h and 12 km/h.

**Figure 6 sensors-25-05407-f006:**
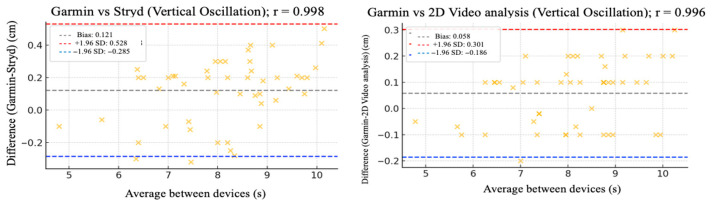
Bland–Altman plots comparing vertical oscillation between the Garmin and the other devices at 9 km/h.

**Figure 7 sensors-25-05407-f007:**
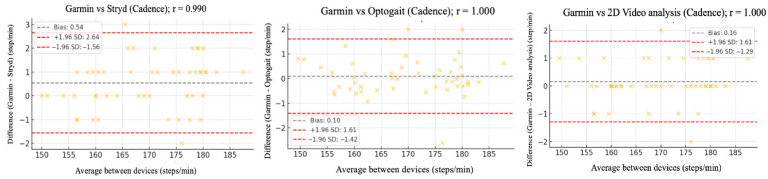
Bland–Altman plots comparing cadence between the Garmin and the other devices at 12 km/h.

**Figure 8 sensors-25-05407-f008:**
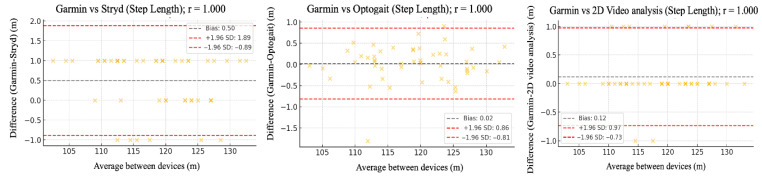
Bland–Altman plots comparing step length between the Garmin and the other devices at 12 km/h.

**Figure 9 sensors-25-05407-f009:**
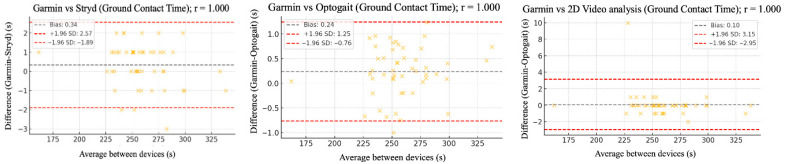
Bland–Altman plots comparing ground contact time between the Garmin and the other devices at 12 km/h.

**Figure 10 sensors-25-05407-f010:**
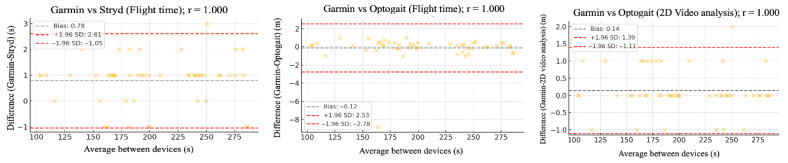
Bland–Altman plots comparing flight time between the Garmin and the other devices at 12 km/h.

**Figure 11 sensors-25-05407-f011:**
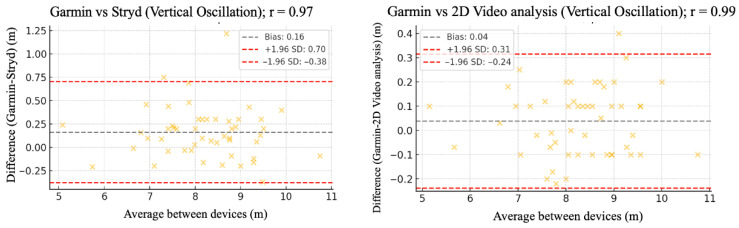
Bland–Altman plots comparing vertical oscillation between the Garmin and the other devices at 12 km/h.

**Table 1 sensors-25-05407-t001:** Spatiotemporal parameters values of each device analyzed at 9 and 12 km/h.

	9 km/h						12 km/h					
	Garmin (a) Mean (SD)	Stryd (b) Mean (SD)	Optogait (c) Mean (SD)	2D Video Analysis (d) Mean (SD)	*p*-Value	Post Hoc	Garmin (a) Mean (SD)	Stryd (b) Mean (SD)	Optogait (c) Mean (SD)	2D Video Analysis (d) Mean (SD)	*p*-Value	Post Hoc
Cadence (spm)	162.10 (8.605)	161.94 (8.326)	162.00 (8.463)	162.22 (8.524)	0.172		169.48 (9.813)	168.94 (9.607)	169.38 (9.821)	169.32 (9.715)	0.009	a > b *; b < c *
Step length (m)	0.947 (0.654)	0.946 (0.658)	0.949 (0.653)	0.947 (0.651)	0.005	b < c *	1.186 (0.071)	1.181 (0.073)	1.186 (0.071)	1.185 (0.071)	0.001	a > b ***; b < c ***; b < d **
Flight time (s)	0.185 (0.543)	0.184 (0.543)	0.185 (0.544)	0.185 (0.543)	0.002	a < b **; b > c **; b > d *	0.204 (0.516)	0.203 (0.516)	0.204 (0.516)	0.204 (0.517)	0.001	a > b ***; b < c ***; b < d ***
Ground Contact time (s)	0.283 (0.238)	0.282 (0.237)	0.282 (0.248)	0.283 (0.239)	0.095		0.259 (0.278)	0.258 (0.280)	0.258 (0.277)	0.258 (0.281)	0.009	a > c *
Vertical oscillation (m)	0.081 (0.012)	0.080 (0.012)		0.080 (0.012)	<0.001	a > b ***; b < d * a > d **	0.083 (0.010)	0.081 (0.010)		0.082 (0.010)	0.001	a > b ***; b < d **

Data are expressed as mean (standard deviation); Spm: step per minute; (* *p* < 0.05; ** *p* < 0.01; *** *p* < 0.001).

**Table 2 sensors-25-05407-t002:** Parameters of relative reliability at both 9 km/h and 12 km/h obtained from the different devices.

		9 km/h					12 km/h				
		r	R^2^	ICC (2,1)	ICC (2,k)	IC (95%)	r	R^2^	ICC (2,1)	ICC (2,k)	IC (95%)
Cadence	Garmin vs. Stryd	0.990	0.981	0.990	0.995	(0.982–0.994)	0.994	0.988	0.992	0.996	(0.989–0.997)
	Garmin vs. Video analysis	0.996	0.992	0.996	0.998	(0.993–0.998)	0.997	0.994	0.997	0.999	(0.995–0.998)
	Garmin vs. Optogait	0.994	0.988	0.994	0.997	(0.989–0.997)	0.997	0.994	0.997	0.998	(0.989–0.997)
Step length	Garmin vs. Stryd	0.990	0.980	0.990	0.995	(0.982–0.994)	0.995	0.991	0.993	0.996	(0.992–0.997)
	Garmin vs. Video analysis	0.997	0.993	0.996	0.998	(0.998–0.999)	0.998	0.996	0.998	0.999	(0.997–0.999)
	Garmin vs. Optogait	0.998	0.995	0.997	0.999	(0.996–0.999)	0.998	0.996	0.998	0.999	(0.997–0.999)
Flight time	Garmin vs. Stryd	1.0	1.0	1.0	1.0	(1.0–1.0)	1.0	1.0	1.0	1.0	(1.0–1.0)
	Garmin vs. Video analysis	1.0	0.999	1.0	1.0	(1.0–1.0)	1.0	1.0	1.0	1.0	(1.0–1.0)
	Garmin vs. Optogait	1.0	1.0	1.0	1.0	(1.0–1.0)	1.0	0.999	1.0	1.0	(0.999–1.0)
Contact time	Garmin vs. Stryd	0.998	0.995	0.997	0.999	(0.996–0.999)	0.999	0.998	1.0	1.0	(0.999–1.0)
	Garmin vs. Video analysis	1.0	0.999	0.997	0.999	(0.996–0.999)	0.999	0.997	0.998	0.999	(0.997–0.999)
	Garmin vs. Optogait	0.951	0.905	0.950	0.975	(0.964–0.985)	1.0	1.0	1.0	1.0	(1.0–1.0)
Vertical Oscillation	Garmin vs. Stryd	0.988	0.976	0.982	0.991	(0.987–0.995)	0.967	0.934	0.956	0.978	(0.942–0.981)
	Garmin vs. Video analysis	0.996	0.992	0.994	0.997	(0.992–0.997)	0.991	0.982	0.991	0.995	(0.984–0.995)

r = correlation coefficient; R^2^ = coefficient of determination; ICC = intraclass correlation coefficient; IC: interval confidence.

## Data Availability

The data that support the findings of this study are available from the corresponding author upon reasonable request.

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
