# Peer review of "Validation of Garmin HRM-Pro for Assessment of Spatiotemporal Parameters During Treadmill Running: Agreement with Three Motion Analysis Systems"

_sensors, 2025, doi:10.3390/s25175407_

Round 1
Reviewer 1 Report
Comments and Suggestions for Authors
General comments
The manuscript aims at measuring the agreement between the Garmin HRM-Pro Band and other validated devices for the computation of spatiotemporal parameters in running.
The study is of interest for this journal. However, I think it is not yet ready for publication on this journal at this stage, as it requires some improvements.
The main concern regards the way results are presented. since the section should include numbers that are basically missing. Of course you put the complete list of results in the Tables, but reporting at least the main numerical results in text will greatly help the reader have the bigger picture of your findings.
More importantly, you should not leave all the Bland-Altman analysis as a supplementary material. It is a fundamental part of your manuscript, as it involves two main characteristics:
- The systematic bias (mean difference).
- The precision of your system (± 1.96 SD of the difference), that gives you a numerical result about how your differences are spread.
It may also give you a third fundamental hint, which is related to the distribution of error (or disagreement in this case), being either homoscedastic or heteroscedastic (i.e., a proportional bias). You may want to give a read to this (10.11613/BM.2015.015). You can find heteroscedastic data visually already in some of your figures, e.g., Fig. 5a or10a, meaning that the difference increases as the measured value increases. This is a sub-optimal situation, as you cannot say that the system is reliable by itself (when compared to the reference one), since it needs a calibration equation that removes the heteroscedasticity.
Moreover, you should state in your Bland-Altman plots how you did compute the difference: reference – your system? Or viceversa? Furthermore, you should add measure units for all of your plots and tables when you refer to every variable.
In the discussion section, I like the way you discussed the literature, since this gives context to your findings. However, I strongly recommend discussing your findings as well, commenting on which systems over/underestimated which other and about what your explanation for that is.
I leave hereinbelow my specific comments, hoping that the authors can find them helpful.
Specific comments
Title: I do not think it is necessary to specify which devices you compared the Garmin with. Conversely, I do think that you should specify in the title that you measured the agreement on treadmill running, since it is not clear until one reads the methods section. I strongly recommend changing the title.
Line 36-37: I would rephrase “including […] state” using expressions like “improving musculoskeletal and cardiovascular health, … improving psychological …. Moreover, what do you mean by “body composition”? Please, expand on this.
Line 50: remove “among others”, or at least place it before the reference [8].
Line 68—70: I would add limited capture volume, which is the key factor that limits (if not completely disrupts) the capability of optical motion capture for long-distance running.
Line 81—82: What is the survey you are talking about? Please, cite it.
Line 95: Why did you specify (e.g., 15 participants)? They may be enough, depending on the kind of study scientists are doing. I would recommend retaining this statement.
Line 98—113: I think that the two paragraphs can be merged into one.
Line 125—126: You should also report the ethics committee approval number.
Line 127: Remove MATERIALS.
Line 129: Body mass, in place of body weight.
Line 130—131: The explanation regarding the BMI computation is superfluous: you can remove it.
Line 138: I would avoid the word “nipples” and use proper anatomical wording. Please, change the language using “in correspondence to the xiphoid process” instead. This stands also for Line 165.
Line 146: You can add the link as a regular citation.
Line 150: I would use the mm as measure unit instead, as per Optical MotionCapture systems like Vicon.
Line 183: I think it would be better for the reader if you change the orientation plane of this picture. It would be way more intuitive if you rotate it either by 45 or 90 degrees (i.e., the participant is running "out of the screen"). In this way, it seems that the camera is placed under the treadmill.
Line 188—190: You did not comment on the normality of the variable distributions. Were they all normal? If not, did you use ANOVA as well? This would not be appropriate for your analysis. Please, expand on this. Moreover, what kind of post-hoc analysis did you perform? Please, expand on this as well.
Line 197: Use the symbol ∙ (central dot) instead of the *.
Line 201: Remove the hyphen from il-lustates.
Line 202—203: This is a mistake. The systematic bias is the mean difference; the CI of the Bland-Altman analysis is ± 1.96 SD of the differences. Rephrase this accordingly.
Line 204: Remove the hyphen from moder-ate.
Line 191—206: Check all the spaces and be consistent with them. E.g., p < 0.05, not p<0.05.
Line 248: The authors, instead of these authors.
Line 253: I think that it would be better for the flow of your discussion if you report these values in Km/h. Just to be consistent and help the reader contextualize these numbers.
Line 262: I would rather use accuracy instead of precision (which is another metric).
Author Response
We would like to thank the time spent reviewing our manuscript. We have considered all suggestions and we believe our manuscript is stronger as a result of the changes that we have introduced in the revised version of the manuscript. Changes to the original manuscript are highlighted in yellow font, and an itemized point-by-point response to your comments in this document.

Reviewer 2 Report
Comments and Suggestions for Authors
- General Comments
This study investigated the validity and reliability of several devices—including Garmin, Stryd, OptoGait, and photogrammetry—for measuring spatiotemporal parameters during running. The paper was generally well written; however, there are several concerns regarding the methodology and the presentation of the results, as outlined below.
- Specific comments
- Major
- The results presented in Table 1 were difficult to interpret, as several parameters showed identical values across devices. For example, the step length at 9 km/h had the same mean (0.95) and standard deviation (0.07) for all devices, yet the reported p-value was 0.005. This suggests either an error in reporting the results or a limitation in the resolution of certain devices. If it is the former, the results should be carefully rechecked. If it is the latter, the conclusion may overestimate the performance of some devices. This concern affects the overall reliability and interpretation of the paper.
- The results presented in Table 2 were also difficult to interpret. Some parameters showed exactly the same values between devices—for example, flight time at 9 km/h was identical between Garmin and OptoGait, as well as between Garmin and photogrammetry. Additionally, some of the reported R² values appear to be incorrect. For instance, one case reported an R value of 0.901, but the corresponding R² was listed as 0.684, which is clearly inconsistent. These issues raise concerns about the accuracy of the analysis and data reporting.
- The authors selected photogrammetry as the gold standard method; however, I believe this choice is not appropriate for estimating spatiotemporal parameters. For kinematic parameters such as step length and vertical oscillation of the center of gravity, a 3D motion capture system should be used as the gold standard. For flight time and ground contact time, force platforms are generally considered the most accurate reference. Furthermore, the calculation methods used in the photogrammetry approach were not described in the paper, making the validity of the reference values debatable.
- Minor
- The authors stated that “The gold standard in kinematic running analysis has been 2D and 3D photogrammetry using high-speed cameras [22]” (P2, L66–68), and “2D video motion analysis has been used previously, showing that it is a valid technology to assess spatiotemporal running metrics [22, 32]” (P4, L161). However, the cited reference [22] is a systematic review comparing photogrammetry with 3D motion capture systems, with the latter being recognized as the true gold standard. Furthermore, this review did not specifically address spatiotemporal parameters. As for reference [32], the study evaluated the validity of two wearable devices for spatiotemporal parameters using photogrammetry as the reference, but it did not validate photogrammetry itself. Therefore, these references are not appropriate to support the authors’ claims, and the validity of using photogrammetry for calculating spatiotemporal parameters remains uncertain.
- There is a misspelling of the abbreviation "CGT" in both the abbreviation section and on page 5, line 210. Please confirm whether this should be "GCT" (ground contact time).
Author Response
We would like to thank the reviewer for the time spent reviewing our manuscript. We have considered all suggestions and we believe our manuscript is stronger as a result of the changes that we have introduced in the revised version of the manuscript. Changes to the original manuscript are highlighted in yellow font, and an itemized point-by-point response to your comments.

Reviewer 3 Report
Comments and Suggestions for Authors
This manuscript examines the synergistic effects of accelerometer, optical system, and high-speed video analyses on spatiotemporal variables during running. It validates the accuracy of spatiotemporal parameters provided by the Garmin HRM-Pro band, which is beneficial for applications in rehabilitation training, running guidance, and balance assessment. The detailed experimental procedures presented in the study enhance its practical utility. Below are specific comments from the reviewers:
- The Garmin HRM-Pro band can be utilized in sports science for assessing the spatiotemporal variables in running, clinical gait analysis, and training optimization. It may be worthwhile to propose a comprehensive evaluation index for comparing the reliability of different devices in gait analysis. This index could assign varying weights to parameters such as ground contact time, step length, flight time, step frequency, and vertical oscillation of the center of mass..
- It is recommended to include control groups with varying speeds to enhance the diversity of the samples.
- Was the reduced sampling frequency of 1Hz for the comparative device detrimental to the accuracy of the original device?
Author Response
We would like to thank the reviewer for the time spent reviewing our manuscript. We have considered all suggestions and we believe our manuscript has improved as a result of the changes that we have introduced in the revised version of the manuscript. Changes to the original manuscript are highlighted in yellow colour, and an itemized point-by-point response to your comments through this document.

Reviewer 4 Report
Comments and Suggestions for Authors
1,Does the introduction provide sufficient background and include all relevant references?
The introduction provides a comprehensive background on the importance of measuring spatiotemporal variables in running, their relevance to performance and injury prevention, and the limitations of existing systems such as photogrammetry and optical systems. However, some more recent references could be included to further strengthen the literature review.
Improvement: Consider adding a few more recent references, especially those published within the last 2-3 years, to ensure the introduction covers the most up-to-date research in this field.
2,Is the research design appropriate?
The research design is appropriate, utilizing a cross-sectional study with 50 recreational runners and comparing the Garmin HRM-Pro band against three different devices (OptoGait, Stryd, and photogrammetry). The use of different speeds (9 and 12 km/h) also helps to evaluate the device's performance under varying conditions.
Improvement: It would be useful to include a power analysis to justify the sample size used. Additionally, considering a follow-up study with a longitudinal design could provide further insights into the device's reliability over time.
3,Are the methods adequately described?
The methods are adequately described, including participant recruitment, inclusion criteria, data collection procedures, and statistical analyses. However, some details regarding the calibration and setup of each device could be further elaborated.
Improvement: Provide more detailed information on how each device was calibrated and set up before data collection. This will help ensure reproducibility of the study.
4,Are the results clearly presented?
The results are clearly presented in tables and text, showing the mean values and standard deviations for each variable measured by the different devices. The statistical significance of differences between devices is also noted.
5,Are the conclusions supported by the results?
The conclusions are well supported by the results, which show that the Garmin HRM-Pro band demonstrates excellent concurrent validity across key biomechanical variables when compared to other devices, especially the gold standard.
Improvement: While the conclusions are generally well supported, the authors could provide more discussion on the practical implications and potential applications of their findings in sports science, clinical gait assessment, and training optimization.
6,Are all figures and tables clear and well-presented?
The figures and tables are clear and well-presented. They are properly labeled and the data is easy to follow.
Comments and Suggestions for Authors
The study is well-designed and the results are promising, demonstrating the potential value of the Garmin HRM-Pro band in measuring spatiotemporal variables during running. However, to strengthen the manuscript, the authors should consider the following:
- Incorporate more recent references in the introduction to ensure a comprehensive literature review.
- Provide additional details on the calibration and setup of each device used in the study.
- Expand the discussion on the practical implications and potential applications of the findings.
Author Response
Dear reviewer,
Thank you so much for giving us the opportunity to improve our manuscript. We have reviewed all your comments carefully and we have added all your suggestion in the new version of the manuscript. You can find the answer to your comment through this document. Also, you can find it included in the manuscript highlights in yellow color.

Round 2
Reviewer 1 Report
Comments and Suggestions for Authors
I would like to thank the authors for their answers to my previous round of comments. The manuscript soundness improved since its original version. However, there are still major concerns about some key aspects that require further attention and editing before the manuscript can be ready for publication. Here are my comments.
---
You specified that you used Kinovea as the software for performing motion-capture analysis. In spite of this, it is erroneous to refer to it as “photogrammetry”, since the latter is the process of overlapping multiple 2D images to reconstruct a 3D model, while you just used a single camera. It is required that you edit the language to match the description of the tool you used. You may want to use “video analysis” or even Kinovea as well.
The ICC analysis should be expanded by considering both the ICC(2,1) and the ICC(2,k). The first one will measure the reliability between individual values computed by different systems; the second one will measure the reliability at rather level, the rather being each subject.
The way results are presented is still not compliant with scientific publications. There is little-to-no numerical result in the text. I strongly encourage you to rewrite them. Moreover, it is impossible for the reader to get any numerical insight from the figures (see contextual comment below). The Pearson’s correlation analysis on the Bland-Altman plots should be reported as well either in a Table or in the text. Numerical results should support each statement in such a section.
- The Bland-Altman plots for FT revealed good agreement between the Garmin HRM-Pro and the three reference systems (bias = XX; LoA = [YY, ZZ]).
- The bias remained small across all comparisons (range = [XX-YY]), with the Garmin–Stryd pairing displaying the narrowest limits of agreement (LoA = [XX-YY]), suggesting high consistency between these two inertial-based devices.
Line 215-216. What did you select as random effects? Please specify this crucial aspect.
Table 1. The second row presents a lot of hyphens that should be avoided. Please consider using acronyms that are later specified in the table’s caption.
Table 2. See my comment for ICC(2,1) and ICC(2,k). Moreover, the caption should specify every acronym appearing in the table. The reader should be able to read it without any other supporting text.
Figures. The resolution of the figures is very low in the pdf I have access to. In any other case than a pdf rendering issue, it is required that you improve the resolution of the pictures to a better quality (e.g., 300 dpi).
Moreover, make sure that all the ticks possess the same number of digits for all the plots.
Author Response
We would like to thank the time spent reviewing our manuscript. We have considered all your comment and we believe our manuscript is stronger as a result of the changes that we have introduced in the revised version of the manuscript. Changes to the original manuscript are highlighted in yellow color, also you can find in this document an itemized point-by-point response to your comments.

Reviewer 2 Report
Comments and Suggestions for Authors
Thank you for the opportunity to review variable report again. The author sincerely revised the manuscript, however, there were several concerns about the references and reliability of description.
- The author revised the manuscript at P2 L66 “analysis has been 3D photogrammetry”. However, the contact time recorded by force plates seemed not included in the 3D photogrammetry. I think it is more appropriate to use the word “3D motion capture system”.
- The author added the statements “Also, this system has limited capture volume, which is the key factor that limits the capability of optical motion capture for long-distance running [23].” However, only 120 seconds motion during treadmill running were recorded in this study. There was no difficulty to record with the 3D motion capture system. Please confirm the validity to use 2D photogrammetry for gold standard in this study. The author showed the table “Reliability of 2D vs. 3D Analysis at Different Frequencies” in the reply comments, the references were not fully described. So, i can't confirm the references. In the revised manuscript, there seemed to be extended interpretations, yet. I referred the validity of the 2D photogrammetric method with Kinovea (not a markerless motion capture system with several cameras) compared to the results of the 3D motion capture system.
- There was a misspelling in P3 L113 “analysis analysis”.
- The author added the description of the sample size calculation. However, the information is insufficient to calculate the required participant’s number. Please provide more details of the calculation process.
- The author added the placement of camera at P4 L170. However, the placement seems to differ from the Figure 1. Which is true?
- Also, the analyzed duration seemed to be inconsistent between P4 L179 and P5 L 201-204. Was the duration analyzed at each procedure differ? Please clarify the analyzed duration in this study.
- The author added the statements about the reliability and validity about the 2D protocol at P4 L182-185, and they used Kinovea in this study. However, the referenced article [37] used Qualysis 3d motion capture system as gold standard and Theia3D markerless motion capture system equipped with 8 cameras was investigated. There was obvious difference between Kinovea (1 camera and 2D result) and Theia3D (8 cameras and 3D result). I think it is extended interpretation, and please check all references throughout the manuscript.
- The figures 2-6 were very poor resolutions. Please make it clear.
Author Response
We would like to thank the reviewer for the time spent reviewing our manuscript. We have considered all your suggestions and we believe our manuscript is stronger as a result of the changes that we have introduced in the revised version of the manuscript. Changes to the original manuscript are highlighted in yellow color, also you can find in this document an itemized point-by-point response to your comments.

Reviewer 3 Report
Comments and Suggestions for Authors
The manuscript has been revised in response to the previous review comments; however, several issues remain:
1、The paper lacks discussion on key confounding factors such as types of running shoes, variability in stride length, and tightness of sensor placement, which may impact the detection of gait parameters. It is recommended to address these factors by standardizing protocols (e.g., using the same shoe model, fixing sensors consistently) in the methodology section or by conducting sensitivity analyses to assess their effects.
2、The study does not emphasize improvements over previous research (e.g., Smith et al., 2022), such as multi-device comparisons and dual-speed validation, nor does it discuss inherent limitations of the devices (e.g., noise in vertical oscillation due to chest-worn sensors). Suggestions include explicitly highlighting technical advancements in the introduction/discussion (e.g., first-time synchronous validation of Stryd+OptoGait+2D video) and analyzing mechanical coupling errors of chest sensors, with a recommendation for future studies to incorporate lower limb IMU validation.
3、The clarity of figures and tables needs improvement.
Author Response
We sincerely thank the reviewer for the time and effort dedicated to evaluating our manuscript. We have carefully considered all your suggestions and believe that the revisions have significantly improved the quality and clarity of our work. All changes made to the manuscript are highlighted in yellow for easy reference. Additionally, we provide below a detailed, point-by-point response to each of your comments
